# Transformation of Residual Açai Fruit (*Euterpe oleracea*) Seeds into Porous Adsorbent for Efficient Removal of 2,4-Dichlorophenoxyacetic Acid Herbicide from Waters

**DOI:** 10.3390/molecules27227781

**Published:** 2022-11-11

**Authors:** Rolando Ramirez, Carlos Eduardo Schnorr, Jordana Georgin, Matias Schadeck Netto, Dison S. P. Franco, Elvis Carissimi, Delmira Wolff, Luis F. O. Silva, Guilherme Luiz Dotto

**Affiliations:** 1Department of Environmental and Sanitary Engineering, Federal University of Santa Maria, Av. Roraima, 1000-7, Santa Maria 97105-900, RS, Brazil; 2Department of Natural and Exact Sciences, Universidad de la Costa, CUC, Calle 58 # 55–66, Barranquilla 080002, Atlántico, Colombia; 3Research Group on Adsorptive and Catalytic Process Engineering (ENGEPAC), Federal University of Santa Maria, Av. Roraima, 1000-7, Santa Maria 97105-900, RS, Brazil

**Keywords:** adsorption, residue, herbicide, activated carbon

## Abstract

Brazil’s production and consumption of açai pulp (Euterpe oleracea) occur on a large scale. Most of the fruit is formed by the pit, which generates countless tons of residual biomass. A new purpose for this biomass, making its consumption highly sustainable, was presented in this study, where activated carbon (AC) was produced with zinc chloride for later use as an adsorbent. AC carbon formed by carbon and with a yield of 28 % was satisfactorily used as an adsorbent in removing the herbicide 2,4-dichlorophenoxyacetic acid (2,4-D). Removal efficiency was due to the highly porous surface (Vp = 0.467 cm^3^ g^−1^; Dp = 1.126 nm) and good surface área (SBET = 920.56 m^2^ g^−1^). The equilibrium data fit the Sips heterogeneous and homogeneous surface model better. It was observed that the increase in temperature favored adsorption, reaching a maximum experimental capacity of 218 mg g^−1^ at 328 K. The thermodynamic behavior indicated a spontaneous, favorable, and endothermic behavior. The magnitude of the enthalpy of adsorption was in agreement with the physical adsorption. Regardless of the herbicide concentration, the adsorbent displayed fast kinetics, reaching equilibrium within 120 min. The linear driving force (LDF) model provided a strong statistical match to the kinetic curves. AC with zinc chloride (ZnCl_2_), created from leftover açai biomass, is a potential alternative as an adsorbent for treating effluents containing 2,4-D.

## 1. Introduction

Modern agriculture aims to ensure a high yield of crops, as this directly impacts the final profit and guarantees the supply of food. In recent years, there has been an indiscriminate increase in the use of herbicides to guarantee these results [1]. Several factors may be involved in this increase. Among them are monocultures and disease resistance over the years. One major problem affecting final production is the presence of undesirable weeds that compete directly with the agricultural crop for macro- and micronutrients in the soil, affecting its development [2]. A herbicide used in large quantities is 2,4-dichlorophenoxyacetic acid (2,4-D) [3,4]. This compound has a low pk_a_ (2.73) [5], so it is found in the anionic form in the pH range existing in the soil, conferring high mobility to the molecule due to its poor retention in the soil [6]. For humans, this compound is toxic and can cause kidney and lung problems, in addition to being considered a carcinogenic and hormonal disruptor potential [7,8]. 

By presenting these characteristics, 2,4-D has great potential for the contamination of surface and groundwater in the environment, alerting the scientific community to develop new techniques for environmental remediation [9,10]. Among the applicable technologies, adsorption has a high potential for use due to its simple design and high efficiency, especially when aligned with porous and high surface area materials such as ACs [11]. In the context of herbicides, several materials have already been used successfully and with high adsorption capacities, such as AC produced from wheat husk to remove 2,4-D [12]; corn cob biochar in the removal of 2,4-D [1]; rice husk hydrochar in the removal of atrazine [13]; bamboo stem biochar in atrazine removal [14]; carbonization of araçá fruit peels to remove atrazine [15]; charcoal production from the bark of the cedar forest species to remove atrazine [16]; removal of diuron from charcoal produced from baobab seeds [17]; and finally, commercial AC was used to remove paraquat, diquat and difenzoquat from the water [18].

When analyzing that the cost of the adsorbent involves most of the expenses in the adsorption process [19], using materials from waste with zero initial cost is a great advantage, in addition to bringing sustainable benefits and being environmentally friendly [20]. The açai (*Euterpe oleraceae*) is a tropical fruit with characteristics similar to the grape; its extraction is carried out in native palm trees where a large part is found in the Amazon, Brazil, being the only income of native communities in the region [21]. Its pulp allows the processing of various products consumed in large quantities around the world [22]. In Brazil, tons are produced annually; only 20% corresponds to the pulp. The rest is generated byproducts, often inappropriately disposed of in the environment [23,24]. Due to these factors, some studies have analyzed the potential of this biomass as an adsorbent in the removal of caffeine [25], heavy metals [26,27,28,29], dyes [30,31], and phenol [32].

This study used the adsorbent developed from açai residues to remove the herbicide 2,4-D. First, whether zinc chloride (ZnCl_2_) is a good activating agent in producing AC was analyzed. The material’s physical properties, such as surface area and pore development, were analyzed in this aspect. Then, it was analyzed if the produced adsorbent had a good adsorption capacity of 2,4-D. The optimal operating parameters, such as temperature and kinetic time influence, were also determined. The experimental data were fitted to isothermal and linear kinetic driving force models. The thermodynamic parameters were determined.

## 2. Results and Discussion

### 2.1. Features of Açai-Derived AC

In this study, the ash content was about 4.7% of the material, and the calculated yield was close to 28%, confirming that most of the residual biomass, about 70%, was converted during the pyrolysis step. This yield is consistent with other studies that carbonized different biomasses with ZnCl_2_ in proportions of 1:1 [5,16,33,34,35]. To analyze the material surface before (Figure 1A,B) and after carbonization with ZnCl_2_ (Figure 1C,D), scanning electron microscopy (SEM) was performed. 

It is possible to observe that there have been major changes on the surface. The original surface was more compact and, in some places, highly smooth, containing some prominences in the form of small circles. When carrying out the carbonization, the surface started to contain new spaces and countless pores distributed randomly and with different sizes and shapes. However, the different shapes and sizes of the particles remained after carbonization. When preparing biochar with açai seed using K_2_CO_3_ as an activating agent in a proportion of 1:3, the authors obtained a surface with the same characteristics as in this study [25]. It should be noted that these new pores can be used as spaces to accommodate the 2,4-D molecules, being highly favorable in the adsorption step.

The results of the two materials’ EDS (energy-dispersive spectroscopy, elemental analysis) (Figure 2) provided data on the proportion of chemical elements on the surfaces. The original substance (Figure 2A) is primarily made of carbon and oxygen, with little amounts of silicon, sodium, and calcium. This composition is consistent with another study that used açai seeds [25]. After carbonization, oxygen and silicon were strongly reduced, and sodium and calcium disappeared completely from the sample. Small proportions of zinc, chlorine, and sulfur were observed in the carbonized sample. They confirmed that the carbonization was carried out successfully, where carbon was >90% of the material. The presence of Zn after washing the material has also been reported in other studies present in the literature [36,37,38]. Due to the substantial loss of volatile material caused by the high temperatures during the pyrolysis process, the composition of the substance generated by carbon is already anticipated [39]. The coal yield and the FT-IR data presented below are compatible with these findings. 

FT-IR was used to identify the primary functional groups on the material’s surface both before and after carbonization, as shown in Figure 3. Both spectra saw weak and strong bands from 3452 to 1034 cm^−1^. Due to the presence of cellulose, lignin, and hemicellulose, as well as water adsorbed on the materials’ surfaces, intense bands were seen in (a), present in both materials. These bands correspond to vibrations of stretching the O-H bond [40,41,42]. The low-intensity band in (b), which is exclusive to the original material, is connected to the lignin’s aromatic, aliphatic, and olefin-specific C-H and C-H_2_ asymmetric stretching vibrations [43,44]. Due to the lignocellulosic material, the bands in (c) present in both materials but with less intensity in AC correspond to aromatic C=C and C=O groups [45]. Additionally, less intense in the AC are the bands in (d), stretching vibrations of the C-O, C-O, and C-O-C bonds of esters, phenols, or alcohols [46]. The activation temperature and the material’s properties are connected to reducing the intensity of particular bands [47]. It was seen that the number of peaks decreased during the pyrolytic process. 

Seeking to understand a little more about the structure of materials, Figure 4 corresponds to the XRD diffractogram of the samples. The original sample shows two noncrystalline peaks attributed to cellulose (II). The peak may be related to organic compounds such as lignin and hemicellulose [48,49,50]. After the carbonization step with ZnCl_2_, the broad, amorphous peak remained; however, a new peak also appeared in the 002 planes of the graphite crystal structure produced during the carbonization process [49]. Therefore, it is confirmed that carbon in its amorphous form was the material’s predominant phase, indicating unorganized structures. This characteristic may favor the adsorption of 2,4-D, since the possible empty spaces are accommodation for the adsorbate molecules. In the literature, it is possible to observe charcoals with similar patterns from different plant biomasses, such as peanut husk [51], Queen palm fruit endocarp [5], Cedrella fissilis husk [16], araucaria bark [52], argan bark [53], Indonesian Kesambi wood [54], and the açai seed carbonized with K_2_CO_3_ [25].

Figure 5a shows the adsorption–desorption isotherms of N_2_ at 77 K, and the pore size distribution is shown in Figure 5b. The large surface area, (S_BET_ = 920.56 m^2^ g^−1^) and the total volume of pores (V_p_ = 0.467 cm^3^ g^−1^) reflect that the carbonization method with ZnCl_2_ was efficient, which was fundamental for the development of pores [55]. Isotherms fall under category I of the IUPAC classification, which corresponds to microporous structures [56,57]. This isotherm agrees with the pore diameter of 1,126 nm, where pores with a diameter of less than 2 nm correspond to microporous structures [58]. Structural characteristics similar to this one agree with other studies using ZnCl_2_ as an activating agent [16,59,60,61].

### 2.2. Effect of Adsorbent Dosage

Figure 6 illustrates how the adsorbent dosage affected the 2,4-D adsorption in AC. The adsorption capacity decreased from 82 to 41 mg g^−1^ when the dosage was increased from 0.4 to 1.2 g L^−1^. On the other hand, the removal increased from 65 to 99%, exhibiting the opposite pattern. The curves intersected at a dosage of 0.6 g L^−1^, which showed satisfactory elimination (95%) and capacity values (79 mg g^−1^). The isothermal and kinetic tests were therefore conducted using an AC dose of 0.6 g L^−1^.

### 2.3. Isothermal and Thermodynamic Studies 

An analysis was done on how the system’s increased temperature affected 2,4-D’s ability to adsorb. As a result, four curves were produced with temperatures ranging from 298 K to 328 K. (Figure 7). Additionally, these analyses clarify the mechanisms governing the interactions between the adsorbent and the adsorbate and enable the calculation of thermodynamic variables. The graphs represent the relationship between the herbicide concentration in the aqueous medium (C_eq_) and the amount of herbicide present on the AC surface (q_eq_).

It is feasible to see that the curves exhibit similar behavior and lack a plateau form, suggesting that there are still open spaces for the adsorbate molecules to occupy. A similar pattern was seen in the 2,4-D adsorption of several ACs made from mushroom remains (*Agaricus bisporus*) [62]. Regarding the adsorption capacity, it was observed that it showed a favorable behavior to the increase in temperature, going from 184 to 218 mg g^−1^. The capacity increased along with the concentration, reaching 218 mg g^−1^ at a concentration of 200 mg L^−1^ and 88 mg g^−1^ at a concentration of 50 mg L^−1^, both of which were measured at 328 K. Additionally, it was discovered that raising the temperature to 328 K produced palm endocarp with a greater 2,4-D adsorption capacity of 367.77 mg g^−1^ [5]. 

Table 1 provides the isothermal data that fit the Langmuir, Freundlich, and Sips models. The Sips isotherm, which had the highest R^2^ and R^2^_adj_ values (>0.9863) and the lowest values ARE (3.169%) and MSR (53.7 (mg g^−1^)^2^), was found to have the most satisfactory results. This model suggests that 2,4-D molecules may be adsorbed on heterogeneous and homogeneous surfaces [12]. K_s_ showed an increase relative to the temperature increase in the system, from 0.32 to 0.50. In the literature, it is possible to observe other studies that showed a better adjustment of herbicide molecules to the Sips isotherm [12,63,64,65,66,67,68].

When comparing the maximum experimental capacity obtained by the AC of 218 mg g^−1^, we saw that it has great application potential compared to other adsorbents in the literature also used in the removal of 2,4-D. When using cotton-derived AC, a capacity of 3.93 mg g^−1^ was observed [69]. When using chitin, the capacity was 6.07 mg g^−1^ and 11.16 mg g^−1^ with chitosan [7]. A capacity of 30.12 mg g^−1^ was obtained using LDH [10]. With magnetic graphene, the capacity was 32.31 mg g^−1^ [70]. When preparing biochar from corn residue, a capacity of 37.4 mg g^−1^ was observed [1]. A nanosized rice hull presented a capacity of 76.92 mg g^−1^ [71]. Carbon nanotubes were also used as adsorbents, obtaining a capacity of 83.33 mg g^−1^ [72]. Poly (glycidyl methacrylate) functionalized with amino acid obtained a capacity of 99.44 mg g^−1^ [73]. Biochar by switch-grass (Panicumvirgatum) had a capacity of 133 mg g^−1^ [74], and ACs from dates [75] and palm endocarp [5] had maximum capacities of 238.1 and 367.77 mg g^−1^, respectively. These comparative results have evidenced that ACs such as the one produced in this study have excellent potential as an adsorbent in the removal of the 2,4-D herbicide.

Table 2 contains a list of the thermodynamic variables examined during the adsorption of 2,4-D in AC. It is possible to determine the type and spontaneity of the adsorption process using thermodynamic parameters [76]. The Ke values climbed from 1.35 to 2.50 as the system temperature increased from 298 to 328 K, indicating that the process is more favorable at a higher temperature. The Gibbs energy decreased from −23.58 to −27.63 kJ mol^−1^, with the greatest negative value at 328 K, showing that the adsorption of 2,4-D in AC was favorable and spontaneous. An endothermic process was confirmed since the enthalpy value (∆H^0^) remained positive (16.16 kJ mol^-1^) and agreed with the isothermal studies. The physical forces are consistent with the magnitude of ∆H^0^ [77]. Thus, it may be concluded that physisorption, which may be reversible and permit desorption, is the main process in our investigation. Furthermore, the significant affinity for 2,4-D molecules on the AC surface is confirmed by the positive value of ∆S^0^ (0.1331 kJ mol^−1^ K^−1^). When analyzing the studies present in the literature, it was observed that many did not analyze the nature of adsorption; however, it is possible to observe that the same behavior was observed by Salomón et al. [5], confirming the endothermic process in herbicide adsorption.

### 2.4. Kinetic Studies and Application of the Linear Driving Force Model (LDF)

The kinetic studies were carried out to analyze the performance in the adsorption of 2,4-D and its equilibrium time at different concentrations. In order to determine the adsorption capabilities at concentrations of 25, 50, and 100 mg L^−1^, the time varied, changingd from 0 to 180 min (Figure 8). However, all curves showed a similar behavior where the capacity increased with the increasing concentration of 2,4-D. Regarding equilibrium, it was observed that it was reached after 120 min with maximum capacities of 37, 79, and 164 mg g^−1^ for concentrations of 25, 50, and 100 mg L^−1^, respectively. Therefore, it can be said that AC has a fast kinetic rate and is favorable for full-scale applications, where a large volume of contaminated effluent can be treated in a shorter period, which generates a lower energy expenditure. However, it is also possible to observe a rapid rate of initial adsorption followed by a slower rate as the adsorption approaches the equilibrium time. This result is expected, given that most of the adsorptive sites on the AC’s surface were free during the initial min and that the rate typically decreases as saturation approaches [74].

Table 3 shows the estimated parameters and statistical indicators for the linear driving force model (LDF). It was found that LDF presented a good fit for the adsorption kinetic data of the AC/2,4-D system. The R^2^ value was above 0.93, and the ARE and MSE values were below 16.78% and 76.52 (mg g^−1^)^2^, respectively. These results showed that the changes were appropriate, and the model can depict the kinetics of 2,4-D adsorption. The model’s estimates of the adsorption capacity, which were 44.81, 83.79, and 163.9 mg g^−1^ for concentrations of 25, 50, and 100 mg L^−1^, respectively, were consistent with those made experimentally. It should be noted that the adsorption rate is higher as the concentration increases, where the driving force exerts an influence between the initial concentration of 2,4-D and the concentration on the surface of the adsorbent [78,79]. In this sense, it was observed that the values of k_LDF_ increased from 7.78 to 31.36×10^−3^ s^−1^ with the increasing concentration. Showing similar behavior, the diffusivity values also increased from 8.49 to 34.23×10^−8^ cm^2^ s^−1^, according to the initial concentration. Additionally, this behavior was noted in studies using ACs to remove phenol [35], 2,4-D [62], and atrazine [16].

## 3. Materials and Methods

### 3.1. Chemicals and Reagents Used

The 2,4-D adsorption studies and the preparation of the carbonaceous material both utilized every product and reagent listed below. Sigma-Aldrich, USA provided the analytical grade zinc chloride (ZnCl_2_), hydrochloric acid (HCl), and 2,4-D (IUPAC name: 2,4-dichlorophenoxy acetic acid, chemical formula: C_8_H_6_C_l2_O_3_, molecular weight: 221.04 g mol^−1^, max = 331 nm, and CAS-Number: 94-75-7). All solutions with different concentrations of 2,4-D were prepared with demineralized water. One gram of 2,4-D was used to create the stock solution, yielding a 1000 mg L^−1^ solution. 

### 3.2. Preparation of AC and Characterization

The source of all açai seeds was a Brazilian factory in Amazonas. The material did not need pulp, because it had already been manually washed with deionized water. After that, 500 g were placed in an oven and dried at 328 K for 48 hours, completely removing the material’s moisture. The kernels were subsequently processed in a knife mill and sieved (using a 250-m sieve) to standardize the particle diameter. Next, the powder was exposed to a 90% ethanol solution for 48 hours to eliminate all extractives. After that, it was cleaned with distilled water and dried for 48 hours in an oven set at 328 K. The remainder, or around 30 g, was utilized to prepare the ACs, while a portion of the material obtained was designated as the precursor material and used for a characterization study. 

Thirty grams of the precursor substance was combined on a plate with thirty grams of zinc chloride (ZnCl_2_) in a ratio of 1:1. Then, 5 mL of distilled water was added, and the mixture was continued until a thick and uniform paste was produced. The paste was dried at 378 K for 48 hours before being macerated in a mortar to create particles smaller than 355 µm. The particles were then put into a quartz tube to perform pyrolysis (under an N_2_ atmosphere with a flow rate of 0.25 L min^−1^; heating rate of 283.15 K min^−1^ to 973.15 K; time of 90 min). 

In the third step, all the pyrolyzed solid material was placed in a solution of HCl (10 mol L^−1^) for 120 min, when ZnCl2 was acid-extracted. After multiple measurements, the pH of the mixture was found to be neutral (pH = 7). The solid components were then separated by decantation and rinsed with distilled water. The material was then crushed to produce particles with a diameter of less than 149 µm after being dried at 323.15 K for 300 min. A tiny portion of the AC from the açai stone was removed for characterization analysis, and the remaining material was employed in 2,4-D batch adsorption tests. The Appendix A provide a demonstration of the characterization process (Appendix A). The estimation of the AC yield (Y, %) according to Equation (1):(1)$$Y(\%)=\left(1\;-\;\frac{m_{f}}{m_{i}}\right)\;\cdot \;100$$
where m_i_ is the initial precursor mass (g), and m_f_ is the precursor mass after pyrolysis. 

### 3.3. Adsorption Experiments

First, dosages of 0.4, 0.6, 0.8, 1, and 1.2 g L^−1^ of AC were introduced separately to Erlenmeyer flasks containing 50 mL of 2,4-D solution at 50 mg L^−1^ to conduct the optimal dosage testing. Next, at room temperature, the samples were shaken for four hours. Then, at the optimal dose of AC, the isothermal and kinetic investigations were carried out (0.6 g L^−1^). Next, the 2,4-D starting concentrations were varied to 0, 50, 100, 150, and 200 mg L^−1^, with all samples prepared in Erlenmeyer flasks and shaken for five hours. The temperatures employed were 298, 308, 318, and 328 K. The final step was to conduct kinetic investigations using 50 mL solutions made in Erlenmeyer flasks. In this instance, the herbicide concentrations were 25, 50, and 100 mg L^−1^ at 298 K, with samples being taken using an aliquot at intervals of 0, 5, 10, 20, 30, 60, and 180 min. 

All of the experiment’s concentrations came from repeated dilutions of this solution. Using a UV–Vis spectrophotometer (Biospectro, Brazil) at 229 nm, all concentration values were measured to preserve the linear relationship between absorbance and concentration. All tests were conducted at the solution’s natural pH for the assays under actual settings (around 6.1). The 2,4-D/AC separation was achieved by centrifuging the samples (Centribio, 80-2B, Brazil) at 4000 rpm for 20 min after each of the adsorption tests described in this paper was carried out in a temperature-controlled shaker (Marconi, MA 093, Brazil) at 180 rpm.

### 3.4. Equilibrium Models and Thermodynamic Parameters

In the experimental data, the following models were chosen: Freundlich [80], Sips [81], and Langmuir [82] to represent the isotherms. The thermodynamic parameters (Gibbs free energy, enthalpy, and entropy) are based on the equilibrium constant of the best-fitted isotherm, in this case, the methodology proposed by Tran [83]. Detailed information regarding the isotherm models and thermodynamic parameters estimation can be found in the Appendix A (Appendix A and Appendix A).

### 3.5. Adsorption Kinetics

The adsorption kinetics is mainly focused on the effect of the adsorbate initial concentration on the time to reach the equilibrium and possible adsorption mechanism. In this work, the linear driving force (LDF) model was selected [84]. Given that Sips was the model that best explained the experimental data, Equation (2) displays the LDF. The Appendix A illustrates the model’s derivation (S5):
(2)$$\frac{d\bar{q}}{\mathrm{dt}}=k_{\mathrm{LDF}}\left(q_{\mathrm{mS}}\frac{K_{S}{\left(C_{0}-D_{0}\bar{q}\right)}^{n_{S}}}{1+K_{S}{\left(C_{0}-D_{0}\bar{q}\right)}^{n_{S}}}-\bar{q}\right)$$
(3)$$\bar{q}\left(t=0\right)=0$$
where n_S_ is the Sips exponent factor (dimensionless), k_LDF_ is the LDF coefficient (s^−1^), K_S_ is the Sips constant (L mg^−1^)^nS^, C_0_ is the initial concentration (mg L^−1^), D_0_ is the adsorbent dosage (g L^−1^), $\bar{q}\;$is the adsorption capacity at any time (mg g^−1^), q_mS_ is the Sips isotherm model parameter, and t is the experiment time (s).

### 3.6. Parameter Estimation, Differential Equation Solution, and Model Evaluation

Matlab scripting was used for the parameter estimates, equation solving, and model evaluation. Therefore, built-in features were used: *particleswarm* was employed for the determination of the parameter’s initial guess, *nlinfit* was employed for the determination of the model parameter without any constraints, *lsqnonlin* was employed for the determination of the parameter with constraints, and *ode15s* was the solver employed for the solution of the LDF model. The equations for each statistical indicator are shown in the Appendix A (Appendix A). The model’s quality fit analysis was achieved by using the statistics parameter determination coefficient (R^2^), adjusted coefficient of determination (R^2^_adj_), average relative error (ARE,%), and minimum squared error (MSE (mg g^−1^)^2^).

## 4. Conclusions

The residual biomass of the açai production chain was successfully carbonized using zinc chloride as an activating agent. The carbonaceous material presented several pores scattered alternately on its surface. The adsorbent showed a good surface area (920.56 m^2^ g^−1^) with a pore diameter of 1.126 nm and pore volume of 0.467 cm^3^ g^−1^. The dosage of 0.6 g L^−1^ showed good values of the capacity and removal. It was found that the system temperature plays an important role, where its increase leads to higher adsorption capacities. The highest capacity (218 mg g^−1^) obtained was 328 K. According to the statistical indicators, the Sips model was the most suitable, confirming a heterogenous and homogeneous surface. The thermodynamic parameters confirmed a process of physical and endothermic nature (∆H^0^ = 16.16 kJ mol^−1^). Regardless of concentration, the adsorption kinetic profiles showed that the process achieved equilibrium in about 120 min. The linear driving force (LDF) model provided a good kinetic representation. Given its good performance, fast kinetics, and high abundance of residual biomass, the material has great potential for use in manufacturing adsorbents with high removal potential.

## Figures and Tables

**Figure 1 molecules-27-07781-f001:**
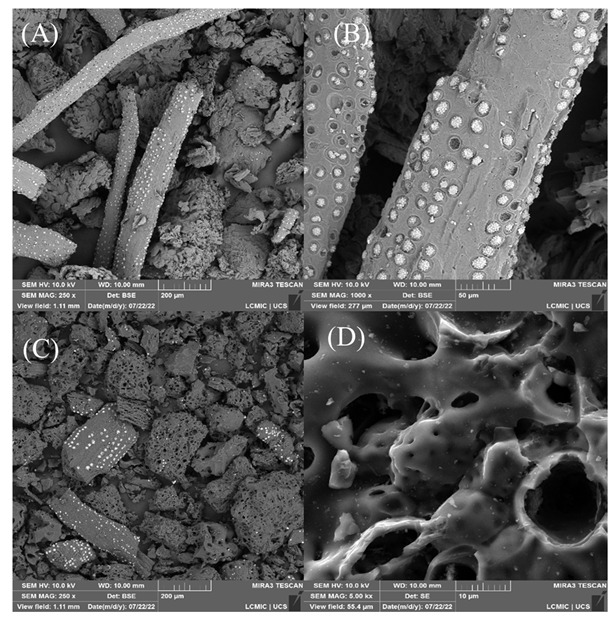
Micrographs of the material surface before (**A**,**B**) and after carbonization (**C**,**D**).

**Figure 2 molecules-27-07781-f002:**
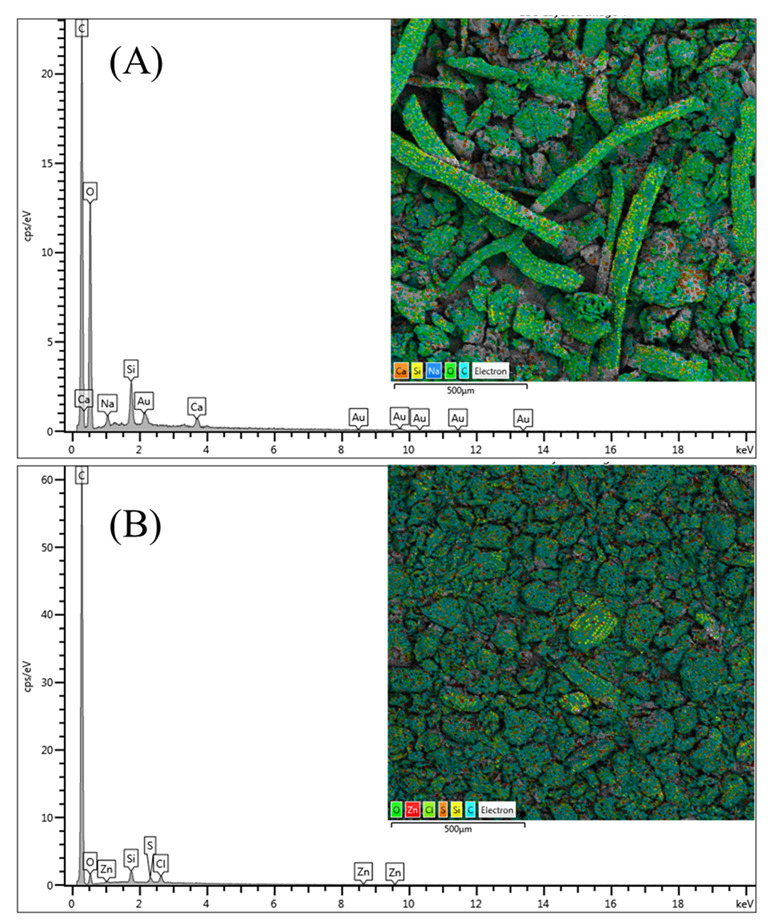
EDS for the precursor (**A**) and AC (**B**).

**Figure 3 molecules-27-07781-f003:**
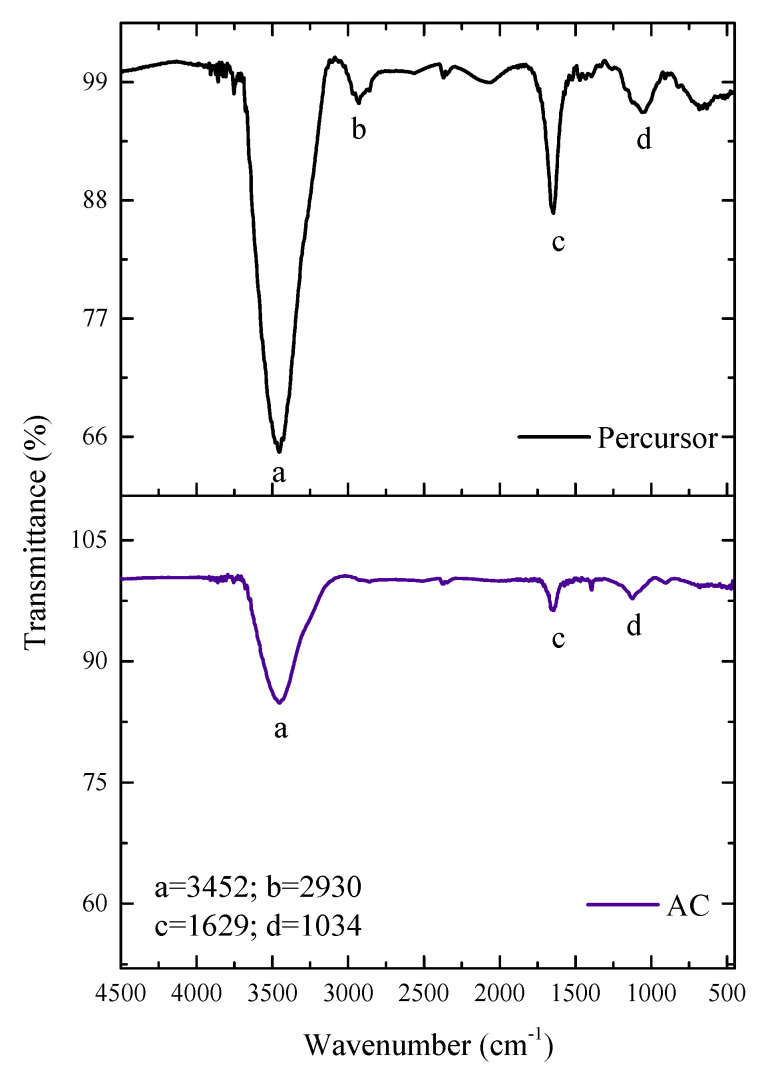
FT–IR spectra of precursor and AC materials.

**Figure 4 molecules-27-07781-f004:**
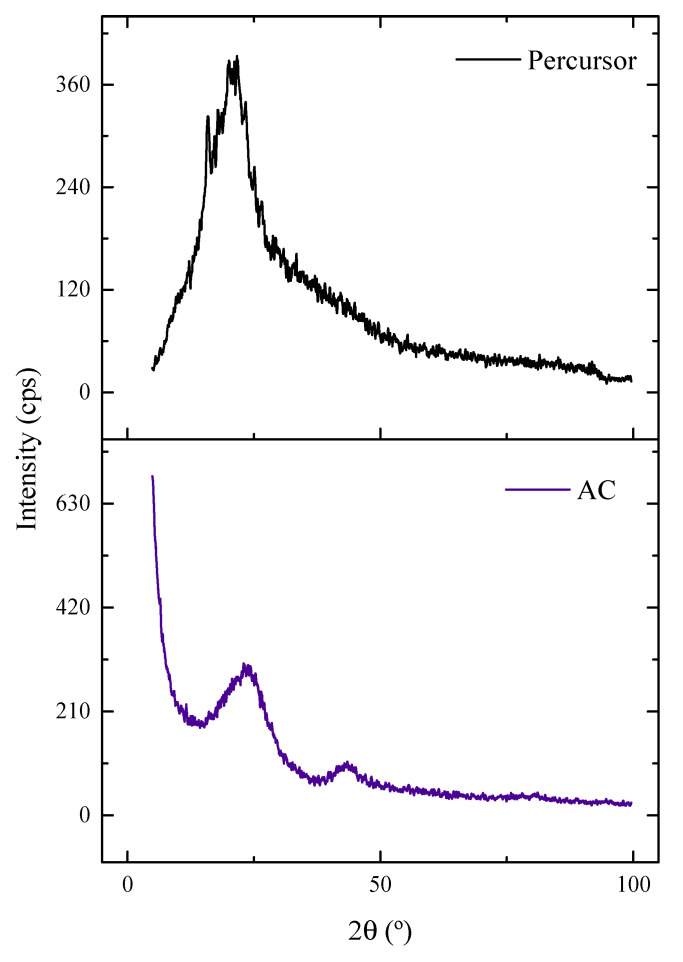
XRD patterns of the precursor material and AC.

**Figure 5 molecules-27-07781-f005:**
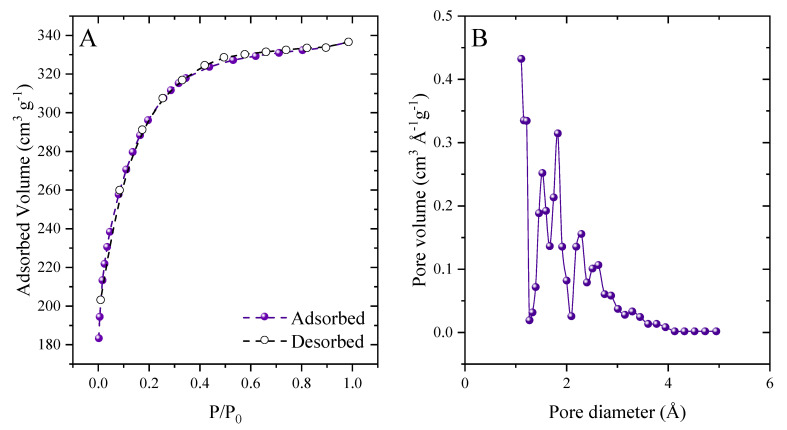
N_2_ adsorption–desorption isotherms (**A**) and desorption pore size distribution (**B**).

**Figure 6 molecules-27-07781-f006:**
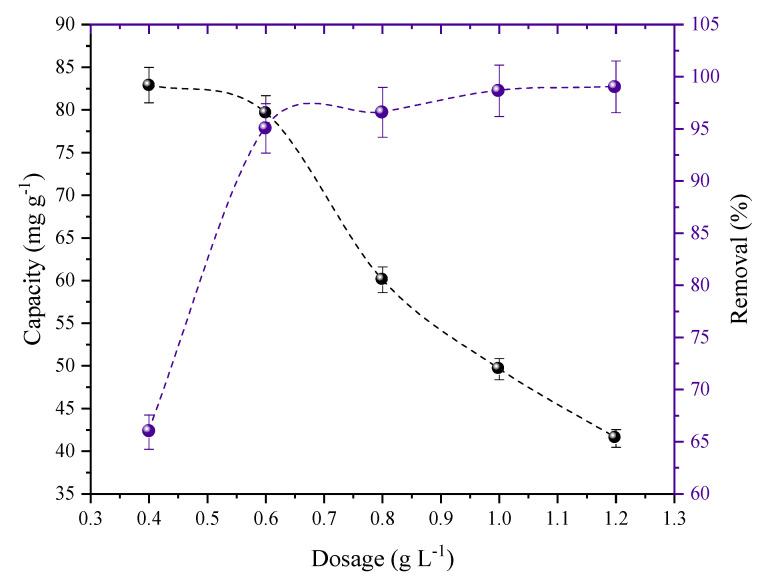
The effects of the AC dosage for the removal and adsorption of 2,4–D (C0 = 50 mg L^−1^; T = 298 K; V = 50 mL; natural pH of the solution; stirring rate = 180 rpm).

**Figure 7 molecules-27-07781-f007:**
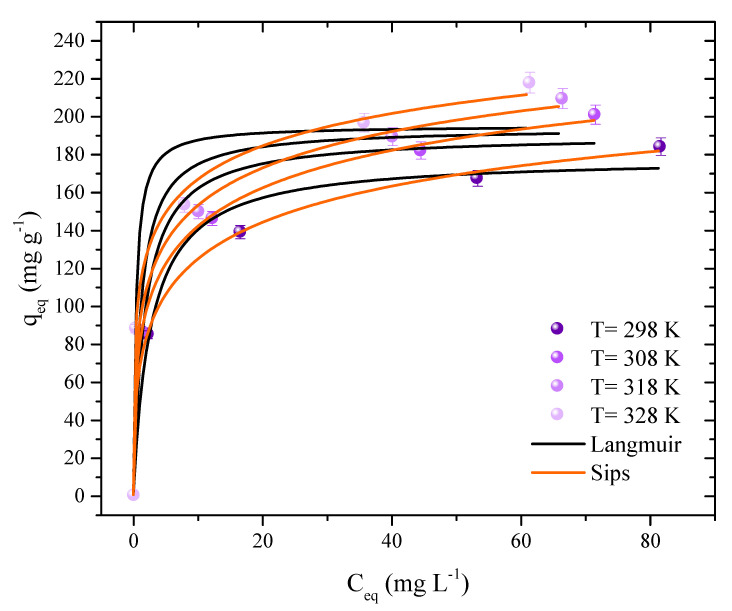
Isothermal adsorption curves of 2,4–D in AC (adsorbent dosage = 0.6 g L^−1^, natural pH of the solution, V = 50 mL and stirring rate = 180 rpm).

**Figure 8 molecules-27-07781-f008:**
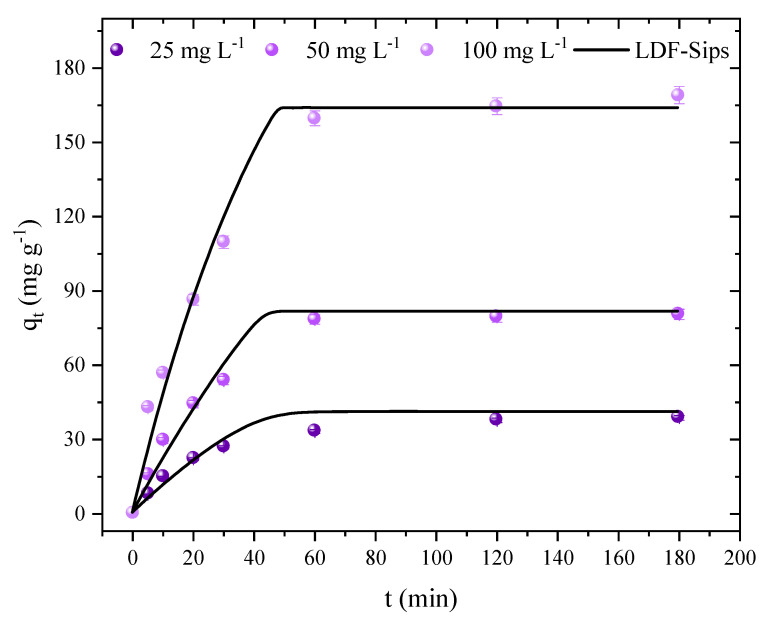
Adsorption capacity at any time (q_t_, mg g^−1^), according to the sample collected; points represent the experimental data and lines the values predicted by the model for the 2,4-D/AC system (T = 298 K, adsorbent dosage = 0.6 g L^−1^, natural pH of the solution, V = 50 mL, and stirring rate = 180 rpm).

**Table 1 molecules-27-07781-t001:** Isothermal parameters for 2,4-D adsorption in AC.

Temperature (K)				
Model	298	308	318	328
Langmuir				
q_mL_ (mg g^−1^)	178.4	190.3	194.1	195.1
K_L_ (L mg^−1^)	0.3657	0.5681	0.8954	2.4473
R^2^	0.9836	0.9740	0.9648	0.9476
R^2^_adj_	0.9673	0.9479	0.9296	0.8952
ARE (%)	6.0592	7.6433	8.6760	9.9616
MSR (mg g^−1^)^2^	121.5	230.9	339.8	548.3
Freundlich				
K_F_ ((mg g^−1^)(mg L^−1^)^−1/nF^)	102.9	113.8	120.1	127.7
1/n_F_ (dimensionless)	0.1136	0.1136	0.1136	0.1136
R^2^	0.9550	0.9534	0.9586	0.9737
R^2^_adj_	0.9100	0.9068	0.9173	0.9474
ARE (%)	11.31	12.48	12.19	10.12
MSR (mg g^−1^)^2^	333.7	413.4	399.2	275.2
Sips				
q_mS_ (mg g^−1^)	276.35	301.68	314.35	327.03
K_S_ (L mg^−1^)^nS^	0.3234	0.3634	0.4098	0.5064
n_S_ (dimensionless)	0.4046	0.3879	0.3636	0.3125
R^2^	0.9993	0.9992	0.9988	0.9966
R^2^_adj_	0.9970	0.9968	0.9950	0.9863
ARE (%)	0.945	1.307	1.867	3.169
MSR (mg g^−1^)^2^	8.242	10.58	17.98	53.70

**Table 2 molecules-27-07781-t002:** Estimated thermodynamic parameters for the adsorption of 2,4-D in AC.

T (K)	K_e_ × 10^−4^	ΔG^0^ (kJ mol^−1^)	ΔH^0^ (kJ mol^−1^)	ΔS^0^ (kJ mol^−1^ K^−1^)
298	1.35	−23.58	16.16	0.1331
308	1.62	−24.84		
318	1.90	−26.06		
328	2.50	−27.63		

**Table 3 molecules-27-07781-t003:** The kinetic parameters were estimated for the adsorption of 2,4-D in the AC.

Model	Initial Concentration (mg L^−1^)		
	25	50	100
LDF-Sips			
q_pred_ (mg g^−1^)	44.81	83.79	163.9
k_LDF_x10^3^ (s^−1^)	7.78	13.61	31.36
D_S_ × 10^8^ (cm^2^ s^−1^)	8.497	14.85	34.23
R^2^	0.9338	0.9795	0.9808
ARE (%)	16.78	14.24	15.03
MSE (mg g^−1^)^2^	13.38	19.72	76.52
q_exp_ (mg g^−1^)	38.79	80.58	169.1

## Data Availability

The data will be available on request.

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
