# Peer review of "Transformation of Residual Açai Fruit (Euterpe oleracea) Seeds into Porous Adsorbent for Efficient Removal of 2,4-Dichlorophenoxyacetic Acid Herbicide from Waters"

_molecules, 2022, doi:10.3390/molecules27227781_

Round 1

Reviewer 1 Report

The manuscript entitled "Transformation of residual açai fruit (Euterpe oleracea) seeds into porous adsorbent for efficient removal of 2,4-dichlorophenoxyacetic herbicide from waters" by Ramirez et.al. mainly used acai pulp to prepare activated carbon to remove 2,4-dichlorophenoxyacetic (2,4-D) herbicide from water. The characteristics of AC were confirmed by SEM, EDS, FTIR, XRD, and BET. The experimental results showed the 82.62% of the simulated effluent. All results also support the conclusion. Therefore, I recommend this manuscript could be accepted after minor revision. 

Some questions for this manuscript are as below.

1. In figure 1B, the magnification should be the same as figure 1D (10 micrometer).

2. Please define q, R, and D0 in Figure 6.

3. In Figure 8, why did all concentrations of AC have the same time to reach saturation of qt?

4. In figure 9, why did the mixture solution contain 2,4-D at a concentration of 50 mg/L and both atrazine and diuron at 10 mg/L to test the simulated effluent?     

Reviewer 2 Report

Major comments: 

1.     Page 2, line 70: The authors write: "No studies were found using açai residues to remove herbicides." Did the authors apply the already known procedure of transformation of residual açai fruit (Euterpe oleracea) seeds into porous adsorbent, or did they develop a new procedure. This cannot be concluded from the manuscript. If it is a new procedure (which significantly increases the quality of the work), it should be emphasized in the last paragraph of the Introduction, but if it is not, a citation is needed in chapter 2.2.

2.     Why was the individual adsorption efficiency of atrazine and diuron not tested? Namely, on the basis of Figure 9, it is not possible to assess whether and how much the mentioned herbicides are adsorbed in addition to 2,4-D, so the text in chapter 3.5 may not be correct. In any case, the text should be reworked. The legend of Figure 9 should also be corrected because it is not "Visible specta".

Minor comments: 

1.     Page 1 of manuscript, line 34: Please check the “….at high production between crops…” formulation.

2.     Page 1, line 41: Authors should provide some citation for pKa.

3.     Page 2, lines 51-58: Please check whether the appropriate references are cited.

4.     In the Introduction (page 2, lines 71-77) the authors should only state the aim of the paper.

5.     Chapter 2 should be chapter 3.

6.     Page 2, lines 87-94: The mentioned text should be moved to a more appropriate place, e.g. after chapter 2.3.

7.     Page 2, line 84: What is “max = 331 nm

8.     Paga 3, line 100: Please check “250 m sieve”.

9.     Paga 3, line 110: Please check “350 m in diameter”.

10.  Paga 3, line 117: Please check “149 m” and so on.

11.  Paga 3, equation (1): “%” should be on the left side of the equation.

12.  Page 4, equations 2 and 2.1: C0 - of which is the initial concentration; what are qms and q with the line above?

13.  Page 5, Figure 1: Letters on figure and in the legend should be the same. The same applies to other figures.

14.  Pages 7 and 8, Figures 3 and 4: Instead of "RAW" it should be written “precursor”.

15.  Page 13, Figure 8: Not defined “qt”.

16.  Decimal separator should be a decimal point instead of a comma in the numbers (e.g., Figures 5a, and 6).

17.  There is no need to introduce the same acronym repeatedly (e.g. 2,4-D). In addition, when an acronym is introduced, it should be used further on in this document. Also, it should be used only after the introduction of acronyms (e.g. 2,4-D; AC, etc.).

18.  Some of the references are not correctly written (4, 9, 10, 12, 14, 31 etc.).

19.  Instead of "percent" you should write "%" (as the authors of the manuscript often do). Similarly, instead of "minute" you should use "min".

20.  In many places a period is placed before and after the citation number. The first point should be omitted.

21.  Missing: Author Contributions, Funding, and Conflicts of Interest.

Round 2

Reviewer 2 Report

With their revised manuscript, the authors have adequately addressed some of the points I made in my review. However, I still have many suggestions for authors. The numbers are as in the previous review. 

Major comments:

1.     Page 2, line 70: The authors write: "No studies were found using açai residues to remove herbicides." Did the authors apply the already known procedure of transformation of residual açai fruit (Euterpe oleracea) seeds into porous adsorbent, or did they develop a new procedure. This cannot be concluded from the manuscript. If it is a new procedure (which significantly increases the quality of the work), it should be emphasized in the last paragraph of the Introduction, but if it is not, a citation is needed in chapter 2.2.

Authors response: Dear Reviewer, the statement has been modified as suggested. 

Reviewer response: The authors modified the statement as suggested. However, in line 71 they mention diuron, not 2,4-D. 

2.     Why was the individual adsorption efficiency of atrazine and diuron not tested? Namely, on the basis of Figure 9, it is not possible to assess whether and how much the mentioned herbicides are adsorbed in addition to 2,4-D, so the text in chapter 3.5 may not be correct. In any case, the text should be reworked. The legend of Figure 9 should also be corrected because it is not "Visible specta".

Authors response: Dear reviewer, the efficiency of the herbicides atrazine and diuron was not the main focus of the study, therefore they were not tested individually. Regarding removal, this is the meta that we use to detect because we don't have a chromatograph. Therefore, this percentage was directed to the total removal of the mixture, not specifying each one of the herbicides individually, as in the other works cited in this paragraph. Therefore, we believe that the statement is correct. The sentence in the figure legend has been corrected to invisible spectra. 

Reviewer response: I think that the conclusions based on this measurement are not reliable, ie. that it is only a rough approximation. The authors do not know which part of the spectra surface of mixture corresponds to which herbicide, therefore, what percentage of each herbicide is adsorbed, so references 66 and 86 should not be used in that sense. Namely, the mentioned three herbicides probably have different molar extinction coefficients, and their concentrations are also different, so that this part of manuscript should be refined. The authors should present the UV-vis spectra of all three the herbicides (individually), as well as in the mixture in order to obtain only qualitative relationships. Also, why is the base line of the spectrum below 0.

Figure 9: Instead of "Unvisible specta" it should read “UV-vis spectra”. It is also necessary to add in figure legend experimental conditions as in the previous figures.

Why was it necessary to add 2.5 times more AC when the substrate increase only about 40%?

Line 173: What mean “….and the pH was measured….”? Maybe set to a specific value?

Minor comments

1.     Page 1 of manuscript, line 34: Please check the “….at high production between crops…” formulation.

Authors response: Dear reviewer, the sentence has been changed.  Modern agriculture aims at high agricultural production, as this directly impacts the final profit and guarantees the supply of food. 

Reviewer response: “…to ensure a high yield of agricultural crops, …” 

2.     Page 1, line 41: Authors should provide some citation for pKa.

Please check reference 5 in reference list. 

5.     Chapter 2 should be chapter 3.

Authors response: Dear reviewer, the authors do not agree with you. Based on other articles online the Introduction should be 1 and Materials and Methods should be 2 as it is. 

Reviewer response: Please see Molecules _ Instructions for Authors_files (https://www.mdpi.com/journal/molecules/instructions) 

7.     Page 2, line 84: What is “max = 331 nm 

Reviewer response: The authors should write "lmax = 331 nm" instead of "max = 331 nm". 

11.  Paga 3, equation (1): “%” should be on the left side of the equation. 

Reviewer response: “%” should be put after Y, ie. "Y (%)" 

12.  Page 4, equations 2 and 2.1: C0 - of which is the initial concentration; what are qms and q with the line above? 

Reviewer response: The authors should write C0 - the initial concentration of 2,4-D in the liquid phase (mg L-1). They should also add in the text below the equations (lines 150-152) what is qms, as well as q with the line above. 

15.  Page 13, Figure 8: Not defined “qt”. 

Reviewer response: It would be better that the authors define "qt" either in the frame of the picture as in Figure 6 (which is not so often), or in the legend of Figure 8 by writing that “qt - capacity as function of the sampling time”. 

17.  There is no need to introduce the same acronym repeatedly (e.g. 2,4-D). In addition, when an acronym is introduced, it should be used further on in this document. Also, it should be used only after the introduction of acronyms (e.g. 2,4-D; AC, etc.).

Authors response: Dear reviewer, the acronym 2,4-D it only used in the introduction. The same for the AC, which is first called in the preparation of activated carbon and characterization section 

Reviewer response: 2,4-D was introduced twice (lines 41 and 84). The acronym 2,4-D needs to be introduced also in the Abstract. The title should also be changed to read: Transformation of residual açai fruit (Euterpe oleracea) seeds into porous adsorbent for efficient removal of 2,4-dichlorophenoxyacetic acid herbicide from waters.

After rearranging the text it is now for AC OK. 

18.   Some of the references are not correctly written (4, 9, 10, 12, 14, 31 etc.).

Authors response: Dear reviewer, the authors did not understand your concern. All the references used were added through Mendeley software and the Molecules citation style was used as a reference. 

Reviewer response:

Reference 4 should be read: Y.-F. Chao, P.-C. Chen, S.-L. Wang, Adsorption of 2,4-D on Mg/Al–NO3 layered double hydroxides with varying layer charge density, Appl. Clay Sci. 40 (2008) 193–200. https://doi.org/https://doi.org/10.1016/j.clay.2007.09.003.

Reference 9: “JS”

Reference 10: Only the first letter of the paper name should be capitalized.

Reference 12: CoCr2O4.

Reference 14: FeCl3.

Reference 31: “LA”.

Authors should also review other references (39 - first author's name; 44, etc.)

Author Response

With their revised manuscript, the authors have adequately addressed some of the points I made in my review. However, I still have many suggestions for authors. The numbers are as in the previous review. 

Major comments:

  1. Page 2, line 70: The authors write: "No studies were found using açai residues to remove herbicides." Did the authors apply the already known procedure of transformation of residual açai fruit (Euterpe oleracea) seeds into porous adsorbent, or did they develop a new procedure. This cannot be concluded from the manuscript. If it is a new procedure (which significantly increases the quality of the work), it should be emphasized in the last paragraph of the Introduction, but if it is not, a citation is needed in chapter 2.2.

Authors response: Dear Reviewer, the statement has been modified as suggested. 

Reviewer response: The authors modified the statement as suggested. However, in line 71 they mention diuron, not 2,4-D. 

Authors response: Dear reviewer, thank you the authors have exchanged the word.

  1. Why was the individual adsorption efficiency of atrazine and diuron not tested? Namely, on the basis of Figure 9, it is not possible to assess whether and how much the mentioned herbicides are adsorbed in addition to 2,4-D, so the text in chapter 3.5 may not be correct. In any case, the text should be reworked. The legend of Figure 9 should also be corrected because it is not "Visible specta".

Authors response: Dear reviewer, the efficiency of the herbicides atrazine and diuron was not the main focus of the study, therefore they were not tested individually. Regarding removal, this is the meta that we use to detect because we don't have a chromatograph. Therefore, this percentage was directed to the total removal of the mixture, not specifying each one of the herbicides individually, as in the other works cited in this paragraph. Therefore, we believe that the statement is correct. The sentence in the figure legend has been corrected to invisible spectra. 

Reviewer response: I think that the conclusions based on this measurement are unreliable, ie. that it is only a rough approximation. The authors do not know which part of the spectra surface of mixture corresponds to which herbicide, therefore, what percentage of each herbicide is adsorbed, so references 66 and 86 should not be used in that sense. Namely, the mentioned three herbicides probably have different molar extinction coefficients, and their concentrations are also different, so that this part of manuscript should be refined. The authors should present the UV-vis spectra of all three the herbicides (individually), as well as in the mixture in order to obtain only qualitative relationships. Also, why is the base line of the spectrum below 0.

Authors response: Dear reviewer, to avoid confusion, this part of the manuscript was removed.

3-Figure 9: Instead of "Unvisible specta" it should read “UV-vis spectra”. It is also necessary to add in figure legend experimental conditions as in the previous figures.

Authors response: Dear reviewer, Figure 9 legend has been updated as follows:

Figure 9. Uv-Vis spectra of the simulated effluent before and after adsorption of the açai residue with charcoal using a 1.5 g L−1 (V= 100 mL, C2,4-D = 50 mg L-1, CAtrazine= 10 mg L-1, CDiuron= 10 mg L‑1, pH=6.1, stirring rate = 150 rpm).

4-Why was it necessary to add 2.5 times more AC when the substrate increase only about 40%?

Authors response: Mainly due to competition reasons. In general, during the adsorption of more chemical species, competition can arise, this is made to ensure better performance. Also, the reviewer should consider that only the adsorbent mass is increased from 0.06 g of adsorbent to 0.15 g, proportional to the higher volume employed.  

5-Line 173: What mean “….and the pH was measured….”? Maybe set to a specific value?

Authors response: Dear reviewer, this was a miss translation the sentence has been updated as follows:

“Then, 1.5 g L−1 of the charcoal prepared in the study was added to the mixture, and the increment of the dosage was made to ensure the satisfactory removal.”

Minor comments

  1. Page 1 of manuscript, line 34: Please check the “….at high production between crops…” formulation.

Authors response: Dear reviewer, the sentence has been changed.  Modern agriculture aims at high agricultural production, as this directly impacts the final profit and guarantees the supply of food. 

Reviewer response: “…to ensure a high yield of agricultural crops, …” 

Authors response: The phrase has been updated as requested: Modern agriculture aims to ensure a high yield of agricultural crops, as this directly impacts the final profit and guarantees the supply of food

  1. Page 1, line 41: Authors should provide some citation for pKa. Please check reference 5 in reference list. 

Reviewer response: Dear reviewer, the authors updated the reference accordingly.

  1. Chapter 2 should be chapter 3.

Authors response: Dear reviewer, the authors do not agree with you. Based on other articles online the Introduction should be 1 and Materials and Methods should be 2 as it is. 

Reviewer response: Please see Molecules _ Instructions for Authors_files (https://www.mdpi.com/journal/molecules/instructions) 

Authors response: Sorry the authors did understand the last question. Now it is clear the that the results should be presented first, thus exchanging the numbering of the sections.

  1. Page 2, line 84: What is “max = 331 nm“ 

Reviewer response: The authors should write "lmax = 331 nm" instead of "max = 331 nm". 

Authors response: The l has been added to the max

  1. Paga 3, equation (1): “%” should be on the left side of the equation. 

Reviewer response: “%” should be put after Y, ie. "Y (%)" 

Authors response: The Eq.1 has been updated

(1)

  1. Page 4, equations 2 and 2.1: C0 - of which is the initial concentration; what are qms and q with the line above? 

Reviewer response: The authors should write C0 - the initial concentration of 2,4-D in the liquid phase (mg L-1). They should also add in the text below the equations (lines 150-152) what is qms, as well as q with the line above. 

Authors response: Dear reviewer, the authors have added all variables to the sentence description of the equation as follows:

Where nS is Sips exponent factor (dimensionless), kLDF is the LDF coefficient (s−1), KS is the Sips constant (L mg−1)nS, C0 is the initial concentration (mg L-1), D0 is the adsorbent dosage (g L−1), is the adsorption capacity at any time (mg g−1), qmS is Sips isotherm model parameter, and t is the experiment time (s). 

  1. Page 13, Figure 8: Not defined “qt”. 

Reviewer response: It would be better that the authors define "qt" either in the frame of the picture as in Figure 6 (which is not so often), or in the legend of Figure 8 by writing that “qt - capacity as function of the sampling time”. 

Authors response: Dear reviewer the authors defined qt in the legends of Figure 8. Also a reminder, all the variables asked for definition are presented in the Supplementary Material!

Figure 8. Adsorption capacity at any time (qt, mg g-1) according to the sample collected, points represent the experimental data and line the values predicted by the model for the 2,4-D/AC system (T = 298 K, adsorbent dosage = 0.6 g L−1, natural pH of the solution, V = 50 mL and stirring rate = 180 rpm).

  1. There is no need to introduce the same acronym repeatedly (e.g. 2,4-D). In addition, when an acronym is introduced, it should be used further on in this document.Also, it should be used only after the introduction of acronyms (e.g. 2,4-D; AC, etc.).

Authors response: Dear reviewer, the acronym 2,4-D it only used in the introduction. The same for the AC, which is first called in the preparation of activated carbon and characterization section 

Reviewer response: 2,4-D was introduced twice (lines 41 and 84). The acronym 2,4-D needs to be introduced also in the Abstract. The title should also be changed to read: Transformation of residual açai fruit (Euterpe oleracea) seeds into porous adsorbent for efficient removal of 2,4-dichlorophenoxyacetic acid herbicide from waters.

After rearranging the text it is now for AC OK. 

Authors response: The title and abstract of the article has been updated as requested. Thank you for the observation.

  1. Some of the references are not correctly written (4, 9, 10, 12, 14, 31 etc.).

Authors response: Dear reviewer, the authors did not understand your concern. All the references used were added through Mendeley software and the Molecules citation style was used as a reference. 

Reviewer response:

Reference 4 should be read: Y.-F. Chao, P.-C. Chen, S.-L. Wang, Adsorption of 2,4-D on Mg/Al–NO3 layered double hydroxides with varying layer charge density, Appl. Clay Sci. 40 (2008) 193–200. https://doi.org/https://doi.org/10.1016/j.clay.2007.09.003.

Reference 9: “JS”

Reference 10: Only the first letter of the paper name should be capitalized.

Reference 12: CoCr2O4.

Reference 14: FeCl3.

Reference 31: “LA”.

Authors should also review other references (39 - first author's name; 44, etc.)

Authors response: Dear reviewer, the authors checked all references. Also, we would like to apologize, the reference style was not well selected in the software.

Round 3

Reviewer 2 Report

Major comments:

1.     Page 2, line 70: The authors write: "No studies were found using açai residues to remove herbicides." Did the authors apply the already known procedure of transformation of residual açai fruit (Euterpe oleracea) seeds into porous adsorbent, or did they develop a new procedure. This cannot be concluded from the manuscript. If it is a new procedure (which significantly increases the quality of the work), it should be emphasized in the last paragraph of the Introduction, but if it is not, a citation is needed in chapter 2.2. 

Authors response: Dear Reviewer, the statement has been modified as suggested. 

Reviewer response: The authors modified the statement as suggested. However, in line 71 they mention diuron, not 2,4-D. 

Authors response: Dear Reviewer, the statement has been modified as suggested.  

Reviewer response: The authors modified the statement as suggested. However, in line 71 they mention diuron, not 2,4-D.  

Authors response: Dear reviewer, thank you the authors have exchanged the word. 

Reviewer response: Unfortunately, diuron is also mentioned in line 72 instead of 2,4-D. 

2.     Authors response: Dear reviewer, the efficiency of the herbicides atrazine and diuron was not the main focus of the study, therefore they were not tested individually. Regarding removal, this is the meta that we use to detect because we don't have a chromatograph. Therefore, this percentage was directed to the total removal of the mixture, not specifying each one of the herbicides individually, as in the other works cited in this paragraph. Therefore, we believe that the statement is correct. The sentence in the figure legend has been corrected to invisible spectra.  

Reviewer response: I think that the conclusions based on this measurement are unreliable, ie. that it is only a rough approximation. The authors do not know which part of the spectra surface of mixture corresponds to which herbicide, therefore, what percentage of each herbicide is adsorbed, so references 66 and 86 should not be used in that sense. Namely, the mentioned three herbicides probably have different molar extinction coefficients, and their concentrations are also different, so that this part of manuscript should be refined. The authors should present the UV-vis spectra of all three the herbicides (individually), as well as in the mixture in order to obtain only qualitative relationships. Also, why is the base line of the spectrum below 0. 

Authors response: Dear reviewer, to avoid confusion, this part of the manuscript was removed. 

Reviewer response: Since the corresponding sections (2.5 and 3.7) were omitted by the authors, this Major comment 2 is no longer actual. However, it is not clear why the authors wrote that the changes in the Figure 9 legend are accepted since that figure is no longer in the manuscript. The same applies to the other authors answers in Major comment 2. 

Minor comments 

7.     Page 2, line 84: What is “max = 331 nm 

Reviewer response: The authors should write "lmax = 331 nm" instead of "max = 331 nm". 

Authors response: The l has been added to the max 

Reviewer response: In the text, now line 269, no change was made. In addition, you should not add "I" before "max", but "l" (lambda). 

17.  There is no need to introduce the same acronym repeatedly (e.g. 2,4-D). In addition, when an acronym is introduced, it should be used further on in this document. Also, it should be used only after the introduction of acronyms (e.g. 2,4-D; AC, etc.).

Authors response: Dear reviewer, the acronym 2,4-D it only used in the introduction. The same for the AC, which is first called in the preparation of activated carbon and characterization section 

Reviewer response: 2,4-D was introduced twice (lines 41 and 84). The acronym 2,4-D needs to be introduced also in the Abstract. The title should also be changed to read: Transformation of residual açai fruit (Euterpe oleracea) seeds into porous adsorbent for efficient removal of 2,4-dichlorophenoxyacetic acid herbicide from waters.

After rearranging the text it is now for AC OK. 

Authors response: After rearranging the text it is now for AC OK. 

Reviewer response: By looking at the manuscript, it can be concluded that the acronym AC is introduced in line 295, while the AC is already mentioned in lines 98, 120, 121, etc.  

18.   Some of the references are not correctly written (4, 9, 10, 12, 14, 31 etc.).

Authors response: Dear reviewer, the authors did not understand your concern. All the references used were added through Mendeley software and the Molecules citation style was used as a reference. 

Reviewer response:

Reference 4 should be read: Y.-F. Chao, P.-C. Chen, S.-L. Wang, Adsorption of 2,4-D on Mg/Al–NO3 layered double hydroxides with varying layer charge density, Appl. Clay Sci. 40 (2008) 193–200. https://doi.org/https://doi.org/10.1016/j.clay.2007.09.003.

Reference 9: “JS”

Reference 10: Only the first letter of the paper name should be capitalized.

Reference 12: CoCr2O4.

Reference 14: FeCl3.

Reference 31: “LA”.

Authors should also review other references (39 - first author's name; 44, etc.) 

Reviewer response: The authors corrected some mistaces, but there are still some (references 10, 39, 44, etc. - only the first letter of the first word of the title of the paper is capitalized, they remained small).

For example: Calisto, J.S.; Pacheco, I.S.; Freitas, L.L.; Santana, L.K.; Fagundes, W.S.; Amaral, F.A.; Canobre, S.C. Adsorption kinetic thermodynamic studies of the 2, 4 – dichlorophenoxyacetate (2,4-D) by the [Co–Al–Cl] layered double hydroxide. Heliyon 2019, 5, e02553, doi:10.1016/j.heliyon.2019.e02553. 399

Author Response

With their revised manuscript, the authors have adequately addressed some of the points I made in my review. However, I still have many suggestions for authors. The numbers are as in the previous review. 

Major comments:

  1. Page 2, line 70: The authors write: "No studies were found using açai residues to remove herbicides." Did the authors apply the already known procedure of transformation of residual açai fruit (Euterpe oleracea) seeds into porous adsorbent, or did they develop a new procedure. This cannot be concluded from the manuscript. If it is a new procedure (which significantly increases the quality of the work), it should be emphasized in the last paragraph of the Introduction, but if it is not, a citation is needed in chapter 2.2.

Authors response: Dear Reviewer, the statement has been modified as suggested. 

Reviewer response: The authors modified the statement as suggested. However, in line 71 they mention diuron, not 2,4-D. 

Authors response: Dear reviewer, thank you the authors have exchanged the word.

Reviewer response: Unfortunately, diuron is also mentioned in line 72 instead of 2,4-D. 

Authors response: Authors are sorry, the diuron in line 72 was replaced with 2,4-D

  1. Why was the individual adsorption efficiency of atrazine and diuron not tested? Namely, on the basis of Figure 9, it is not possible to assess whether and how much the mentioned herbicides are adsorbed in addition to 2,4-D, so the text in chapter 3.5 may not be correct. In any case, the text should be reworked. The legend of Figure 9 should also be corrected because it is not "Visible specta".

Authors response: Dear reviewer, the efficiency of the herbicides atrazine and diuron was not the main focus of the study, therefore they were not tested individually. Regarding removal, this is the meta that we use to detect because we don't have a chromatograph. Therefore, this percentage was directed to the total removal of the mixture, not specifying each one of the herbicides individually, as in the other works cited in this paragraph. Therefore, we believe that the statement is correct. The sentence in the figure legend has been corrected to invisible spectra. 

Reviewer response: I think that the conclusions based on this measurement are not reliable, ie. that it is only a rough approximation. The authors do not know which part of the spectra surface of mixture corresponds to which herbicide, therefore, what percentage of each herbicide is adsorbed, so references 66 and 86 should not be used in that sense. Namely, the mentioned three herbicides probably have different molar extinction coefficients, and their concentrations are also different, so that this part of manuscript should be refined. The authors should present the UV-vis spectra of all three the herbicides (individually), as well as in the mixture in order to obtain only qualitative relationships. Also, why is the base line of the spectrum below 0.

Authors response: Dear reviewer, to avoid confusion, this part of the manuscript was removed. 

Reviewer response: Since the corresponding sections (2.5 and 3.7) were omitted by the authors, this Major comment 2 is no longer actual. However, it is not clear why the authors wrote that the changes in the Figure 9 legend are accepted since that figure is no longer in the manuscript. The same applies to the other authors answers in Major comment 2. 

Authors response Dear reviewer, the correction of the manuscript has been conducted for more the one author. That said, it was forgotten to be removed the answer regarding Figure 9.

Minor comments

  1. Page 2, line 84: What is “max = 331 nm“ 

Reviewer response: The authors should write "lmax = 331 nm" instead of "max = 331 nm". 

Authors response: The l has been added to the max

Reviewer response: In the text, now line 269, no change was made. In addition, you should not add "I" before "max", but "l" (lambda). 

Authors response: The symbol λ (lambda) has been added to the manuscript as requested

  1. There is no need to introduce the same acronym repeatedly (e.g. 2,4-D). In addition, when an acronym is introduced, it should be used further on in this document.Also, it should be used only after the introduction of acronyms (e.g. 2,4-D; AC, etc.).

Authors response: Dear reviewer, the acronym 2,4-D it only used in the introduction. The same for the AC, which is first called in the preparation of activated carbon and characterization section 

Reviewer response: 2,4-D was introduced twice (lines 41 and 84). The acronym 2,4-D needs to be introduced also in the Abstract. The title should also be changed to read: Transformation of residual açai fruit (Euterpe oleracea) seeds into porous adsorbent for efficient removal of 2,4-dichlorophenoxyacetic acid herbicide from waters.

After rearranging the text it is now for AC OK. 

Authors response: The title and abstract of the article has been updated as requested. Thank you for the observation.

Reviewer response: By looking at the manuscript, it can be concluded that the acronym AC is introduced in line 295, while the AC is already mentioned in lines 98, 120, 121, etc.

Authors response: Dear reviewer, thank you for pointing it out, AC is now first called in line 16.

  1. Some of the references are not correctly written (4, 9, 10, 12, 14, 31 etc.).

Authors response: Dear reviewer, the authors did not understand your concern. All the references used were added through Mendeley software and the Molecules citation style was used as a reference. 

Reviewer response:

Reference 4 should be read: Y.-F. Chao, P.-C. Chen, S.-L. Wang, Adsorption of 2,4-D on Mg/Al–NO3 layered double hydroxides with varying layer charge density, Appl. Clay Sci. 40 (2008) 193–200. https://doi.org/https://doi.org/10.1016/j.clay.2007.09.003.

Reference 9: “JS”

Reference 10: Only the first letter of the paper name should be capitalized.

Reference 12: CoCr2O4.

Reference 14: FeCl3.

Reference 31: “LA”.

Authors should also review other references (39 - first author's name; 44, etc.)

Authors response: Dear reviewer, the authors checked all references. Also, we would like to apologize, the reference style was not well selected in the software.

Reviewer response: The authors corrected some mistaces, but there are still some (references 10, 39, 44, etc. - only the first letter of the first word of the title of the paper is capitalized, they remained small).

For example: Calisto, J.S.; Pacheco, I.S.; Freitas, L.L.; Santana, L.K.; Fagundes, W.S.; Amaral, F.A.; Canobre, S.C. Adsorption kinetic thermodynamic studies of the 2, 4 – dichlorophenoxyacetate (2,4-D) by the [Co–Al–Cl] layered double hydroxide. Heliyon 2019, 5, e02553, doi:10.1016/j.heliyon.2019.e02553. 399

Authors response: Dear reviewer, the authors have exchanged the reference style once again, in this case only the first letter of the article name is capitalized

  1. Calisto, J. S.; Pacheco, I. S.; Freitas, L. L.; Santana, L. K.; Fagundes, W. S.; Amaral, F. A.; Canobre, S. C. Adsorption kinetic and thermodynamic studies of the 2, 4 – dichlorophenoxyacetate (2,4-D) by the [Co–Al–Cl] layered double hydroxide. Heliyon 2019, 5, e02553.
  2. Mohd Din, A. T.; Hameed, B. H.; Ahmad, A. L. Batch adsorption of phenol onto physiochemical-activated coconut shell. J. Hazard. Mater. 2009, 161, 1522–1529.
  3. Li, W.; Mo, W.; Kang, C.; Zhang, M.; Meng, M.; Chen, M. Adsorption of nitrate from aqueous solution onto modified cassava (Manihot esculenta) straw. Ecol. Chem. Eng. S 2012, 19, 629–638.